# Feasibility of post-exposure-prophylaxis with single-dose rifampicin and identification of high prevalent clusters in villages' hyperendemic for leprosy in Senegal

Gilbert Batista[1]*, Pauline Dioussé[2], Papa Mamadou Diagne[3], Mahamat Cissé[4], Ibrahima Tito Tamba[1], Fatou Diop[5], Fanta Diop[6], Lahla Fall[7], Diama Sakho[8], Mariama Bammo[9], Ndiaga Guèye[9], Christa Kasang[10], Nimer Ortuño-Gutiérrez[11], Louis Hyacinthe Zoubi[12], Madoky Magatte Diop[2]

1 Action Damien, Dakar, Sénégal, 2 Unité de Formation et Recherche en Santé, Université de Thiès, Sénégal, 3 Association Sénégalaise de lutte contre la lèpre et les MTN (ASCL/MTN), Dakar, Sénégal, 4 Association allemande de lutte contre la lèpre et la tuberculose–DAHW Sénégal, 5 Centre hospitalier régional de Kaolack, Sénégal, 6 Centre hospitalier régional de Louga, Sénégal, 7 Centre hospitalier de l'ordre de Malte–CHOM Dakar, Sénégal, 8 Centre hospitalier régional de Ziguinchor, Sénégal, 9 Centre hospitalier régional de Thiès, Sénégal, 10 DAHW German Leprosy and Tuberculosis Relief Association, Germany, 11 Damien Foundation, Brussels, Belgium, 12 National Leprosy Control Program, Dakar, Sénégal

* drgbatista@gmail.com

**Data Availability Statement:** The data supporting the findings of this publication are retained at Damien Foundation, Belgium, and will not be made

## Abstract

### Introduction

Senegal is a leprosy low-endemic country with nine villages known to be hyperendemic with a leprosy incidence rate above 1,000 per million inhabitants. We aim to implement a door-to-door screening strategy associated with the administration of a single-dose-rifampicin (SDR) as post-exposure prophylaxis (PEP) to household and social contacts in these villages and to identify spatial clustering and assess the risk of leprosy in population according to the physical distance to the nearest index-case.

### Methods

From October/2020 to February/2022 active door-to-door screening for leprosy was conducted in nine villages. Using an open-source application, we recorded screening results, demographic and geographic coordinate's data. Using Poisson model we analysed clustering and estimated risk of contracting leprosy in contacts according to the distance to the nearest new leprosy patient.

### Results

In nine villages, among 9086 contacts listed, we examined 7115. Among 6554 eligible contacts, 97.8% took SDR. We found 39(0.64%) new leprosy cases among 6,124 examined in six villages. Among new cases, 21(53.8%) were children, 10(25.6%) were multibacillary and 05(12.8%) had grade 2 disability. The prevalent risk ratio and 95% confidence intervale (95%CI) adjusted by village were 4.2(95%CI 1.7–10.1), 0.97(95%CI 0.2–4.4), 0.87(95%CI

openly accessible due to ethical and privacy concerns. However, data can be made available after approval of a motivated and written to MTU@damiaanactie.be.

**Funding:** The study was funded by the Damien Foundation Belgium and the German Association for Leprosy and Tuberculosis Relief (DAHW). The funders played no role in the study design, data collection and analysis, decision to publish or preparation of the manuscript. However, some investigators, specially GB, ITT and NO-G are employees of the Damien Foundation; MC and CK are employees of the German Leprosy and Tuberculosis Relief Association -DAHW.

**Competing interests:** The authors have declared that no competing interests exist.

0.2–25), 0.89(95%CI 0.3–2.6) and 0.70(95%CI 0.2–2.5) for the contacts living in the same household of an index case, 1-25m, 26-50m, 51-75m and 76-100m compared to those living at more than 100m respectively. We identified nine high prevalent clusters including 27/39 (69%) of new cases in 490/7,850(6%) inhabitants, with relative risks of 46.6(p-value = 0.01), and 7.3, 42.8, 8.2, 12.5, 11.4, 23.5, 22.3, and 14.6 (non-significant p-values).

## Conclusions

Our strategy has proved the feasibility of active screening for leprosy in contacts and the introduction of PEP for leprosy under programmatic conditions. Only individuals living in the same household as the leprosy patient had a significant risk of contracting leprosy. We documented nine clusters of leprosy that could benefit from tailored control activities while optimizing resources.

## Author summary

Senegal has nine Social Rehabilitation Villages (VRS) which were established in March 1976. These are former leprosaria converted into villages and are known for their high leprosy transmission. To improve screening and reduce transmission of the disease, we implemented from October 2020 to February 2022, an active leprosy screening strategy associated with the administration of a single dose of rifampicin (SDR) as post-exposure prophylaxis (PEP) to household and social contacts in these VRS. This operational research study showed that the strategy was feasible under programmatic conditions with 7115(98.6%) people accepting the test among 7219 present and 6408(97.8%) people accepting and taking SDR among 6554 eligible. We found 39 new cases in six VRS among 6,124 examined. The geographical coordinates of the households were also entered using the adapted 'Open Data Kit' (ODK) application, which allowed the identification of nine highly prevalent clusters including 27/39 (69%) of new cases in 490/7,850 (6%) inhabitants in five VRS with cases detected. Our results can be used to guide targeted and more efficient active case finding.

## Introduction

Leprosy is a chronic complex infectious disease that still affects over 200,000 new cases per year globally [1]. Although the *Mycobacterium leprae* was discovered 150 years ago [2], no reliable tests are available and current diagnosis is mainly based on clinical skills and microscopy [3]. The more recently discovered *Mycobacterium lepromatosis* [4] adds another layer to the complexity of the control of leprosy. Delay of leprosy diagnosis causes permanent disability and is the main cause of stigma and discrimination [5]. Leprosy is classified by the World Health Organization (WHO) among the twenty priority neglected tropical diseases (NTDs [6]. Like other of the NTDs, it is often linked to poor socio-economic conditions.

The current Global Leprosy Strategy 2021–2030 relies on early diagnosis and includes the provision of single-dose rifampicin as post-exposure prophylaxis [7]. A study documented the need that all countries including those with low prevalence should implement SDR-PEP to achieve zero transmission of *Mycobacterium leprae* in 2023 [8].

In Senegal, over the past decade, the nationwide average number of new cases per year was 15.4 per million population with 10.6 in 2020 [9]. Currently, leprosy is under-detected in Senegal, which is confirmed by the detection of several new cases during the active screening campaigns conducted in recent years. The high proportion of multi-bacillary (MB) cases and cases with visible grade 2 disability (G2D), which are 66% and 22% respectively in 2021, are indicators of delay of diagnosis. Although leprosy has not been a public health problem since 1995, Senegal is experiencing an increase in new cases in some areas [10,11]. Thus, in 2016, the National Leprosy Control Program (NLCP) conducted with its partners, a study of leprosy prevalence in two of the nine Social Rehabilitation Villages (SRVs) of Koutal and Mballing that were created in 1976 for the isolation of leprosy cases. This study highlighted a resurgence of leprosy in the two villages with the number of new cases detected in Mballing increasing from 31 cases in 2015 to 63 cases in 2016. It also showed that the incidence of leprosy was significantly higher in families with a history of leprosy and household contacts of leprosy patients represent a population at high risk of contracting the disease [12].

Passive case detection and treatment alone have proven insufficient to interrupt transmission of *Mycobacterium leprae*. To boost the prevention of leprosy, with the consent of the index case, WHO recommends systematic tracing household contacts along with neighbours and social contacts of each index case, accompanied by the provision of SDR-PEP [13] To be cost-effective, active case detection by door-to-door visit of households of the population at risk requires mapping to identify groups with high endemicity. This not only reduces cost but also helps on training of health staff and community sensitization.

In this study, we aimed to implement a door-to-door active screening strategy associated with the administration of SDR-PEP in nine SRVs villages of Senegal. Also, we identified clustering of leprosy cases including the population at risk according to the physical distance to an index case in order to better targeting the beneficiaries for the provision of SDR-PEP.

## Methods

### Ethics statement

The ethical clearance from the National Ethics Committee for Health Research of the Ministry of Health and Social Action (MSAS) was obtained on March 24, 2020 (N˚00000051 MSAS/DPRS/CNERS), as well as the administrative authorization of the MSAS on April 01, 2020 (N˚00000341/MSAS/DPRS/DR).

Written consent was obtained from each participant prior to inclusion in the study, after information on the study objective and procedures had been provided in the local language by the head of each team. For subjects under 18 years of age, written consent was obtained from a legal representative.

### Study design

This was a cross-sectional study. From October 2020 to February 2022, active door-to-door leprosy screening combined with recording of GPS-coordinates of households screened using the ODK (Open Data Kit) application for data entry was conducted in the nine SRVs in Senegal. All consenting household members were screened for leprosy.

### Study setting

The study was conducted in the nine SRVs villages that exist in Senegal. In 2020, the first year of the research project we included the villages of Mballing (Thies Region) and Koutal (Kaolack Region). In 2021, we included the villages of Touba Peycouck (Thies Region), Fadiga

(Kédougou Region) and Sowane (Fatick Region). Finally, in the 2022, the villages of Teubi and Djibélor (Ziguinchor Region), Kolda (Kolda Region), and Diambo Soubalo (Saint-Louis Region) were visited.

Teams composed of health workers (dermatologists, and nurses), community health workers, and people affected by leprosy visited each household in the nine VRS after sensitization of the local leaders, authorities and the community. All residents who signed a consent form were screened for signs of leprosy. Negative household and social contacts aged two years or older were treated with single-dose rifampicin with a dosage of 10 mg/kg. The exclusion criteria were age below two years, pregnancy, allergy to rifampicin, liver and kidney disease, presumptive TB, presumptive or confirmed leprosy, and refusal to take SDR-PEP.

All adverse were recorded and were managed at no cost to the patient according to national pharmacovigilance guidelines in Senegal. The Ministry of Health pharmacovigilance form was used to report adverse events. Among contacts, all newly detected leprosy cases were treated with Multidrug Therapy (MDT) according to the standard WHO protocol. Suspected cases for which slit-skin-smear (SSS) was required to confirm the diagnosis were sampled for this analysis.

### Sample size

In each village, the survey included the estimated population in Mballing of 5160 inhabitants, Koutal with 1100 inhabitants, Teubi with 215 inhabitants, Djibélor with 300 inhabitants, Touba Peycouck with 3500 inhabitants, Diambo Soubalo with 91, Sowane with 500, Kolda with 227, and Fadiga with 980.

### Data collection

The information was collected in a strictly standardized approach. On the one hand, from an interview questionnaire and, on the other hand, from clinical examination for leprosy screening. The data were coded and entered in real time using the ODK open-source application. This allowed collection of information about household and social contact screening, demographic data, results of screening, household geographic coordinates and administration of rifampicin. After verification, the data collected with ODK were sent to a password protected server accessible only to the principal investigators.

### Data analysis

The data entered with the ODK application were analyzed with Epi-Info 6 and STATA software (Version 11, StataCorp, USA) both descriptively and analytically: frequency distribution according to several criteria (age, sex, degree of contact, etc.) and percentage comparisons by the $X^2$ test (chi$^2$) or Fisher's test (one or two-sided) with a consensual risk of 0.05 and a precision of 0.02.

For measuring the distance from the contacts to the nearest index case we used the distance matrix function of Quantum Geographical Information System (QGIS) software [14].

Clusters were analyzed using the Poisson model using Kulldorff's spatial scan statistic with 50% of the population as the maximum cluster size [15].

In Stata, using the Poisson model, we estimated the risk of contracting leprosy among contacts as a function of distance in meters to the nearest new leprosy patient.

## Results

### Coverage of the screening of population

A total of 9086 household members, including 446 former cases of leprosy, were listed. Among household members, 7219(80%) were present at the moment of the survey. The number of

**Table 1.  Coverage of the screening of population in nine VRS of Senegal, 2020–2022.**

| VRS | Household members listed | Present | | Accepting examination | | New leprosy Cases | |
|---|---|---|---|---|---|---|---|
| | N | n | % | n | % | n | % |
| Koutal | 1095 | 854 | 78,0 | 846 | 99,1 | 13 | 1,5 |
| Mballing | 3804 | 3079 | 80,9 | 3039 | 98,7 | 18 | 0,6 |
| Peycouck | 2294 | 1739 | 75,8 | 1709 | 98,3 | 1 | 0,1 |
| Fadiga | 835 | 684 | 81,9 | 670 | 98,0 | 0 | 0,0 |
| Sowane | 323 | 249 | 77,1 | 249 | 100,0 | 0 | 0,0 |
| Teubi | 161 | 132 | 82,0 | 128 | 97,0 | 2 | 1,6 |
| Djibélor | 291 | 236 | 81,1 | 233 | 98,7 | 1 | 0,4 |
| Kolda | 205 | 174 | 84,9 | 169 | 97,1 | 4 | 2,4 |
| Diambo | 78 | 72 | 92,3 | 72 | 100,0 | 0 | 0,0 |
| **Total** | **9086** | **7219** | **79,5** | **7115** | **98,6** | **39** | **0,5** |

household members accepting to be examined was 7115 (99%) and of those examined, 368 (5.2%) were index cases, 2360 (33.2%) were household contacts and 4387 (61.6%) were social contacts living in a household close to that of an index case. In some households, we found several former cases of leprosy but only one was considered an index case, the first to develop the disease. We found 39(0.64%) new leprosy cases among 6,124 examined among 7,850 listed in six villages. (Table 1)

## Characteristics of new leprosy cases

A total of 39 new cases were detected in 6 villages. Excluding the three VRS with no new cases (Sowane, Fadiga, and Diambo), a population of 6124 was examined, which equates to a prevalence of 6.4 per 1000 population (95% confidence interval 5–9).

Among the 39 new cases, 25.6% were multibacillary leprosy (MB) cases, 53.8% were children under 15 years of age, 61.5% were female and 12.8% were cases with grade 2 disability (G2D). (Table 2)

## Characteristics of people who accepted and took the single dose of rifampicin

Of the 7115 household members screened, 561 (7.9%) were excluded from the SDR because they were ineligible. A total of 6408 (97.8%) of the 6554 eligible people took a single dose of

**Table 2.  New leprosy cases detected after active screening in nine Social Rehabilitation Villages of Senegal, 2020–2022.**

| VRS | NC | MB | | <15 years | | Female | | G2D | |
|---|---|---|---|---|---|---|---|---|---|
| | n | n | % | n | % | n | % | n | % |
| Koutal | 13 | 2 | 15,4 | 9 | 69,2 | 9 | 69,2 | 0 | 0,0 |
| Mballing | 18 | 4 | 22,2 | 9 | 50,0 | 12 | 66,7 | 3 | 16,7 |
| Peycouck | 1 | 1 | 100,0 | 0 | 0,0 | 1 | 100,0 | 0 | 0,0 |
| Fadiga | 0 | 0 | 0,0 | 0 | 0,0 | 0 | 0,0 | 0 | 0,0 |
| Sowane | 0 | 0 | 0,0 | 0 | 0,0 | 0 | 0,0 | 0 | 0,0 |
| Teubi | 2 | 1 | 50,0 | 1 | 50,0 | 0 | 0,0 | 0 | 0,0 |
| Djibélor | 1 | 0 | 0,0 | 1 | 100,0 | 1 | 100,0 | 0 | 0,0 |
| Kolda | 4 | 2 | 50,0 | 1 | 25,0 | 1 | 25,0 | 2 | 50,0 |
| Diambo | 0 | 0 | 0,0 | 0 | 0,0 | 0 | 0,0 | 0 | 0,0 |
| **Total** | **39** | **10** | **25,6** | **21** | **53,8** | **24** | **61,5** | **5** | **12,8** |

**Table 3. Distribution of people who accepted and took the single dose of rifampicin in nine Social Rehabilitation Villages of Senegal, 2020–2022.**

| VRS | Eligible | SDR | | 15 years + | | 10–14 years | | 5–9 years | | 2–4 years | |
|---|---|---|---|---|---|---|---|---|---|---|---|
| | n | n | % | n | % | n | % | n | % | n | % |
| Diambo | 69 | 69 | 100,0 | 46 | 66,7 | 4 | 5,8 | 13 | 18,8 | 6 | 8,7 |
| Djibélor | 204 | 202 | 99,0 | 125 | 61,9 | 13 | 6,4 | 43 | 21,3 | 21 | 10,4 |
| Fadiga | 636 | 627 | 98,6 | 395 | 63,0 | 38 | 6,1 | 129 | 20,6 | 65 | 10,4 |
| Kolda | 145 | 144 | 99,3 | 99 | 68,8 | 15 | 10,4 | 21 | 14,6 | 9 | 6,3 |
| Koutal | 721 | 686 | 95,1 | 396 | 57,7 | 58 | 8,5 | 124 | 18,1 | 108 | 15,7 |
| Mballing | 2795 | 2752 | 98,5 | 1703 | 61,9 | 269 | 9,8 | 433 | 15,7 | 347 | 12,6 |
| Peycouck | 1633 | 1580 | 96,8 | 1060 | 67,1 | 117 | 7,4 | 232 | 14,7 | 171 | 10,8 |
| Sowane | 246 | 245 | 99,6 | 174 | 71,0 | 14 | 5,7 | 40 | 16,3 | 17 | 6,9 |
| Teubi | 105 | 103 | 98,1 | 71 | 68,9 | 4 | 3,9 | 16 | 15,5 | 12 | 11,7 |
| **TOTAL** | **6554** | **6408** | **97,8** | **4069** | **63,5** | **532** | **8,3** | **1051** | **16,4** | **756** | **11,8** |

rifampicin as post-exposure prophylaxis. Of these, 63.5% were aged 15 or over, followed by children aged 5 to 9 (16.4%) and 2 to 4 (11.8%). (Table 3). No serious adverse events were reported.

## Probability of being a leprosy patient as a function of distance to nearest index case

We included the SRVs where there were new cases detected. By measuring the distance between a contact and the nearest index case, we created six categories, namely: 0 meters (same household), less than 25 meters, 25–50 meters, 50–75 meters, 75–100 meters and beyond 100 meters. Keeping the same categories we estimated the risk of getting leprosy according to the distance.

Proportion of new leprosy cases is higher in households vs. beyond. As a matter of fact, the prevalent risk ratio and 95% confidence interval(95%CI) adjusted by village in the same household, 1-25m, 26-50m, 51-75m, 76-100m was respectively 4.2(95%CI 1.7–10.1), 0.97(95%CI 0.2–4.4), 0.87(95%CI 0.2–25), 0.89(95%CI 0.3–2.6) and 0.70(95%CI 0.2–2.5) compared to those living at more than 100m. Only individuals living in the same household as the leprosy patient had a significant risk of contracting leprosy. (Table 4)

## Characteristics of clusters identified

Again, for the spatial analysis to identify clustering, we included only SRVs with new cases detected except for Peycouck and Teubi because only one new case has been detected. We identified nine highly prevalent clusters in four VRS including 24/39 (62%) of new cases in 313/5,265 (6%) inhabitants.

**Table 4. Probability of being a leprosy patient as a function of distance to nearest index case random-effects model controlling for village of residence.**

| Distance to index case | Population screened | Number of leprosy cases (%) | Adjusted prevalence rate ratio (95% CI) |
|---|---|---|---|
| Same household | 619 | 12 (1.94) | **4.18 (1.74–10.04)** |
| Neighbours contact at <25 m | 411 | 2 (0.49) | 0.96 (0.21–4.45) |
| Neighbourhood contact at 25–<50 m | 1166 | 5 (0.43) | 0.87 (0.29–2.56) |
| Neighbourhood contact at 50–<75 m | 1116 | 5 (0.45) | 0.89 (0.30–2.64) |
| Neighbourhood contact at 75–<100 m | 799 | 3 (0.38) | 0.70 (0.19–2.56) |
| Neighbourhood contact at >100 m | 3739 | 12 (0.32) | Ref. |

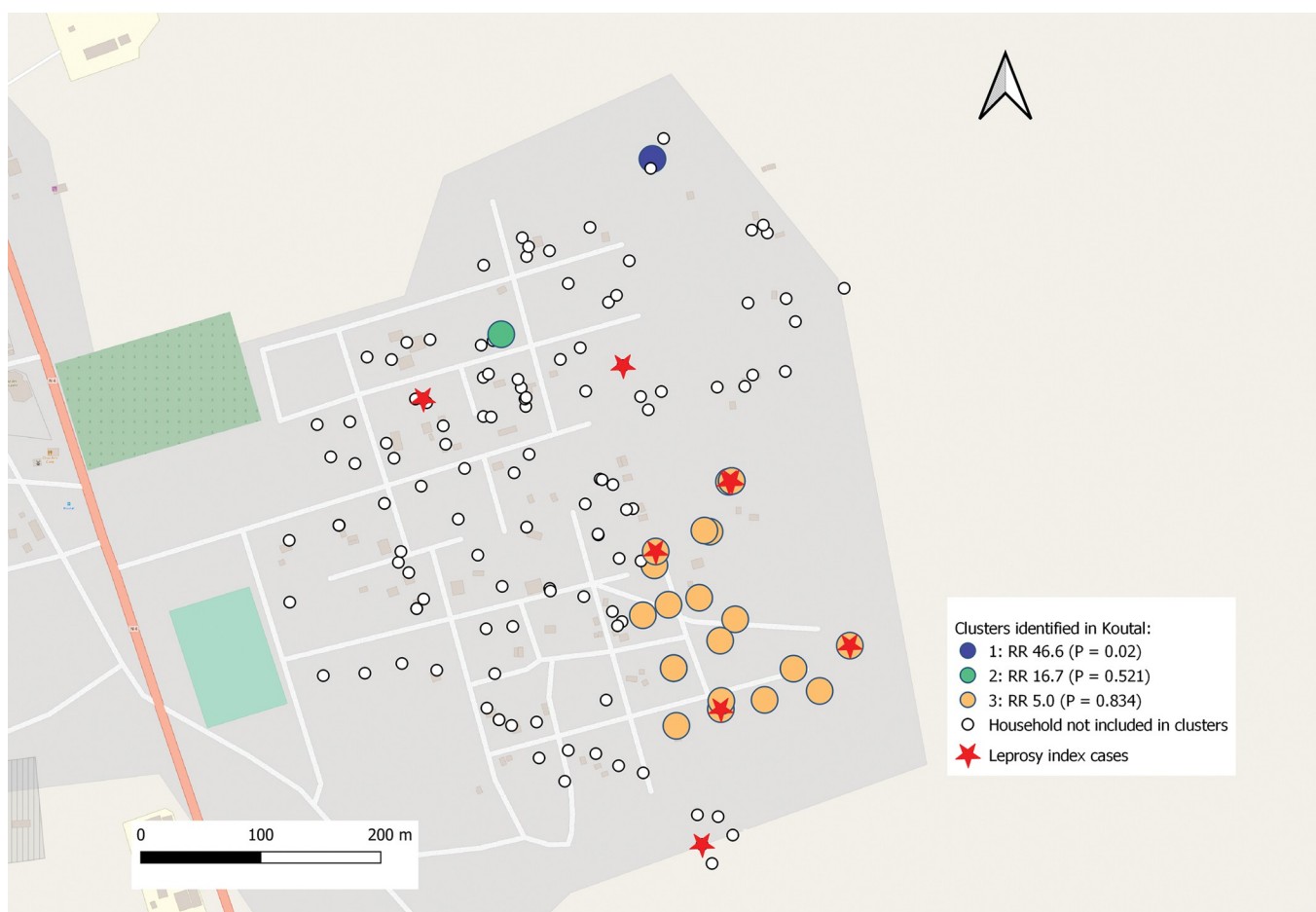

**Fig 1. Clusters identified in Koutal, 2020.** OpenStreet Map was used as a base layer for developing this map (**https://www.openstreetmap.org/#map=13/14.0877/-16.0931**).

The SRV Koutal had three clusters with one of them including two households and seven persons with three cases with a relative risk (RR) of 46.6 and a P-value of 0.02 (Fig 1).

Although clusters were identified in other SRV but there RR was not statistically significant. (Table 5). Indeed, in Mballing, no cluster was statistically significant, but three high transmission clusters were present, including a household with 18 members with an RR of 26. A non-

**Table 5. Characteristics of clusters identified in four villages of Senegal, 2020–2022.**

| VRS | cluster | Number of household | cluster size (population) | Number of leprosy cases | Relative risk | p-value |
|---|---|---|---|---|---|---|
| Koutal | 1 | 1 | 7 | 3 | 46.6 | **0.02** |
|  | 2 | 1 | 10 | 2 | 16.7 | 0.521 |
|  | 3 | 19 | 137 | 5 | 5.04 | 0.834 |
| Mballing | 1 | 14 | 94 | 4 | 11.3 | 0.362 |
|  | 2 | 8 | 57 | 3 | 11.1 | 0.55 |
|  | 3 | 1 | 18 | 2 | 26.3 | 0.6 |
| Teubi | 1 | 8 | 37 | 2 | 5.3 | 0.092 |
| Kolda | 1 | 1 | 3 | 2 | 5.8 | 0.68 |
|  | 2 | 11 | 34 | 2 | 2.4 | 0.71 |

statistically significant cluster was identified in Teubi with 8 households and two cases among 37 members and since the P-value is borderline, a potential risk of transmission still exists in these households. Two non-statistically significant clusters were found in Kolda, including one cluster with one household, three members, and two cases.

## Discussion

We screened 7115 household members in 9 SRV and found 39(0.64%) new leprosy cases in six villages among 6,121 examined, equivalent to a prevalence rate of 64 per 10.000 (95% confidence interval 50–90). These results show a high prevalence of leprosy in these villages and are comparable to the high yield from active leprosy case-finding in Cambodia, where the endemic rate is relatively low, as in Senegal. Indeed, A. Cavaliero et al. found 33 (0.44%) new cases of leprosy in 7496 people screened [16].

The proportion of children among new cases was 54%, this illustrates that there is a recent transmission and therefore need to repeat the survey within two years. Three SRVs had no new cases, a camp approach could be foreseen if no cases are detected in between.

All new cases were detected in households where there are former cases. The proportion of new cases is higher in households vs. beyond. Only individuals living in the same household as the leprosy patient had a significant risk of contracting leprosy, compared to those living at more than 100m [IRR of 4,18 (CI 95% 1.74–10.04)]. This statistically significant association with physical distance from the nearest index case has been found in other countries. In the Comoros, the study by Ortuno-Gutierrez et al showed a higher relative risk of 7.5 [17]. In Brazil, the cohort study by Teixeira CSS et al. found that household contacts of patients with multibacillary leprosy had a higher risk of developing leprosy (adjusted odds ratio [OR], 1.48; 95% CI, 1.17–1.88).[18].

A total of 6408 (97.8%) of the 6554 eligible people took a single dose of rifampicin as post-exposure prophylaxis and no serious adverse event were reported. Post-exposure prophylaxis with SDR was safe and was well accepted by eligible index patients and their contacts. Similar results have been found in many other countries [19]. These results show that our strategy of active leprosy screening combined with the administration of a single dose of rifampicin as post-exposure prophylaxis in VRS is feasible under programmatic conditions. However, it would be desirable to evaluate it in the context of non-VRS, where leprosy is often unrecognized.

We identified nine highly prevalent clusters in four villages. A cluster with a statistically significant RR of 46.6 is present in Koutal, with two others non-statistically significant indicating a high risk of transmission. Regardless of statistically signification, those clusters need to be screened again by door-to-door as the transmission is high and include small numbers of population at risk.

Our results illustrate that leprosy clusters at the household level and that active screening of household contacts is highly efficient. Given the spatial distribution of new cases in clusters and the recent transmission, door-to-door screening is ideally recommended for the next survey.

Despite sensitization campaigns, 20% of the inhabitants listed were absent and were not examined and a qualitative survey can guide about the reasons to be addressed while sensitization of community.

This absence rate and the uncontrollable mobility of populations justify further research in non-endemic areas known as "out-of-area". In addition, stigmatisation plays an important role in the number of unreported cases and refusals of PEP. Overall, there remain subtle but real socio-anthropological factors that make leprosy a persistent neglected tropical disease. This should be integrated into any new approach.

## Conclusion

We found a high yield of active household contact screening. This pilot demonstrated an increased risk of leprosy in contacts in the household.

The SDR-PEP project proved the feasibility of strengthened contact tracing, skin screening, introducing leprosy PEP with SDR and collection of high-quality individual data. Integration of SDR distribution into a leprosy control programme invigorated through the current leprosy control efforts by an increased motivation resulting from the addition of a novel tool and better training.

## Acknowledgments

To the National Leprosy Control Programme, Damien Foundation, German Leprosy and Tuberculosis Relief Association-DAHW, National Association for the Fight against Leprosy and NTDs (ASCL/MTN), Regional Hospitals of Thiès, Louga, Kaolack and Ziguinchor, Ordre de Malte health centre (CHOM).

## Author Contributions

**Conceptualization:** Gilbert Batista, Pauline Dioussé, Papa Mamadou Diagne, Christa Kasang, Nimer Ortuño-Gutiérrez, Louis Hyacinthe Zoubi, Madoky Magatte Diop.

**Data curation:** Gilbert Batista, Nimer Ortuño-Gutiérrez.

**Formal analysis:** Gilbert Batista, Nimer Ortuño-Gutiérrez.

**Funding acquisition:** Gilbert Batista, Mahamat Cissé, Christa Kasang, Nimer Ortuño-Gutiérrez.

**Investigation:** Gilbert Batista, Pauline Dioussé, Papa Mamadou Diagne, Mahamat Cissé, Ibrahima Tito Tamba, Fatou Diop, Fanta Diop, Lahla Fall, Diama Sakho, Mariama Bammo, Ndiaga Guèye, Christa Kasang, Nimer Ortuño-Gutiérrez, Louis Hyacinthe Zoubi, Madoky Magatte Diop.

**Methodology:** Gilbert Batista, Pauline Dioussé, Ibrahima Tito Tamba, Christa Kasang, Nimer Ortuño-Gutiérrez, Madoky Magatte Diop.

**Project administration:** Gilbert Batista, Mahamat Cissé, Louis Hyacinthe Zoubi.

**Resources:** Christa Kasang.

**Software:** Gilbert Batista, Nimer Ortuño-Gutiérrez.

**Supervision:** Gilbert Batista, Pauline Dioussé, Papa Mamadou Diagne, Mahamat Cissé, Louis Hyacinthe Zoubi, Madoky Magatte Diop.

**Validation:** Gilbert Batista, Nimer Ortuño-Gutiérrez.

**Visualization:** Gilbert Batista.

**Writing – original draft:** Gilbert Batista.

**Writing – review & editing:** Gilbert Batista, Nimer Ortuño-Gutiérrez.

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
