## [Decision Letter · Decision Letter 0]

22 Nov 2023

Dear Dr. Batista,

Thank you very much for submitting your manuscript "Feasibility of post-exposure-prophylaxis with single-dose rifampicin and identification of high prevalent clusters in villages’ hyperendemic for leprosy in Senegal" for consideration at PLOS Neglected Tropical Diseases. As with all papers reviewed by the journal, your manuscript was reviewed by members of the editorial board and by several independent reviewers. In light of the reviews (below this email), we would like to invite the resubmission of a significantly-revised version that takes into account the reviewers' comments. 

We cannot make any decision about publication until we have seen the revised manuscript and your response to the reviewers' comments. Your revised manuscript is also likely to be sent to reviewers for further evaluation.

Sincerely,

Joseph M. Vinetz

Section Editor

Joseph Vinetz

Section Editor

Reviewer's Responses to Questions

**Key Review Criteria Required for Acceptance?**

**Methods**

-Are the objectives of the study clearly articulated with a clear testable hypothesis stated?

-Is the study design appropriate to address the stated objectives?

-Is the population clearly described and appropriate for the hypothesis being tested?

-Is the sample size sufficient to ensure adequate power to address the hypothesis being tested?

-Were correct statistical analysis used to support conclusions?

-Are there concerns about ethical or regulatory requirements being met?

Reviewer #1: The methods are fine

Reviewer #2: The objectives of the study are clearly articulated and are well align with the design of the study. The population choosen for the study is appropriate for the implementation of the conducted research. Also the sample size is sufficient enough to ensure the significance of the results. Some terms used in the methodology section are not very clear. When the authors talked about listing or examining the contacts of the leprosy index cases, it was often not clear, are they talking about the family / household contacts or social contacts or neighbours. Each time contacts are mentioned the group should specified. It would have been helpful for the readers if the method section would cleary mention how many leprosy index cases were taken into consideration when designing the intervention. Also how many contacts per each leprosy index case were planned to be screened.

**Results**

-Does the analysis presented match the analysis plan?

-Are the results clearly and completely presented?

-Are the figures (Tables, Images) of sufficient quality for clarity?

Reviewer #1: Results presentation is confusing - see comments.

Reviewer #2: The analysis or the data presented is well align with the methodology or the analysis plan. However, there are some inconsistencies in the numbers that mentioned throughout the paper. For example, the authors mentioned at one point that 6,121 persons were examined / screened and another time the number was mentioned as 6,124. In addition to that, if only 6,121 or 6,124 contacts were examined, how come that 6,554 were eligible and 6,408 were given SDR-PEP. 

Furthermore, in the result section, it was also not always clear how the numbers were calculated, as sometimes only absolute numbers or percentages were given. However, these do not add up when recalculating. The number of contacts who were listed is also not consistent throughout the manuscript. The number of household members (9081) listed and the number of contacts who are listed (7850) are a bit confusing.

**Conclusions**

-Are the conclusions supported by the data presented?

-Are the limitations of analysis clearly described?

-Do the authors discuss how these data can be helpful to advance our understanding of the topic under study?

-Is public health relevance addressed?

Reviewer #1: Conclusions are not supported by the data presented. See comments

Reviewer #2: The conclusion is supported by the data presented. Limitations are also clearly described. The manuscript clearly illustrates the risk of getting leprosy among the household members of the leprosy cases which is very relevant, especially for the low-endemic settings. However, it was not clear why the camp approach if no cases were detected among the household and the risk to other groups is even lower?

**Editorial and Data Presentation Modifications?**

Reviewer #1: review and copy-editing is needed to eliminate typos and correct sentences that are incomplete/do not make sense.

Reviewer #2: (No Response)

**Summary and General Comments**

Reviewer #1: General: Batista and co-authors present in this manuscript the findings of a campaign consisting of active leprosy case detection and post-exposure prophylaxis in Senegalese villages that emerged from former leprosy colonies. They report a high number of new cases with epidemiological characteristics indicating diagnosis delays and recent transmission, and high acceptance of prophylactic treatment. New cases were clustered, yet the authors report no elevated risk beyond the household of index cases, which is counter-intuitive. 

While this is a relevant study for leprosy control program design, the manuscript suffers from a number of inaccuracies and shortcomings which need to be addressed before it can be considered for acceptance.

- Abstract: the results section is confusing – re-organize with only one topic in one sentence (e.g. the first sentence which reports new cases, screening rates and SDR acceptance)

- Abstract: The reported number of individuals screened is lower than the reported number of individuals who received SDR. Since screening is a precondition for SDR this is not possible

- Abstract and Results: Conclusion: “Only individuals living in the same household as the leprosy patient had a significant risk of contracting leprosy.” This is clearly not correct (see table 4) – rather, people outside the household of the index patient had a risk similar as the general population, which was still high based on the reported figures (note: most new cases were reported outside the index case households!). Thus, also the statement “All new cases were detected in households where there are former cases (index cases)” is incorrect. 

- Were the leprosy villages created in 1976 or 1979?

- Results: the stated number of 9081 household members seems implausible given the overall size of the villages. Also, household members should not be called social contacts. Why were not all contacts listed? The following figures are also odd: 7115 accepting to be examined, 6124 examined, 7850 listed – all these figures do not add up and do not make sense. 

- Check for typos (e.g. intervale -> interval, M. Leprae -> M. leprae, SDR-EP -> SDR-PEP)

Reviewer #2: The paper is very relevant and the methodology is clearly articulated. This helps other researchers to replicate the study in different settings. The study is also very useful to understand the risk of getting a leprosy infection among the household contacts of the index cases. Please, recheck the numbers used in the methods and results section. Sometimes they do not add up. Having a clear description of the contact groups and numbers would be very helpful.

PLOS authors have the option to publish the peer review history of their article (what does this mean?). If published, this will include your full peer review and any attached files.

Reviewer #1: No

Reviewer #2: No
---

## [Decision Letter · Decision Letter 1]

14 Jan 2024

Dear Dr. Batista,

We are pleased to inform you that your manuscript 'Feasibility of post-exposure-prophylaxis with single-dose rifampicin and identification of high prevalent clusters in villages’ hyperendemic for leprosy in Senegal' has been provisionally accepted for publication in PLOS Neglected Tropical Diseases.

Best regards,

Stuart D. Blacksell

Section Editor

Joseph Vinetz

Section Editor

Reviewer's Responses to Questions

**Key Review Criteria Required for Acceptance?**

**Methods**

-Are the objectives of the study clearly articulated with a clear testable hypothesis stated?

-Is the study design appropriate to address the stated objectives?

-Is the population clearly described and appropriate for the hypothesis being tested?

-Is the sample size sufficient to ensure adequate power to address the hypothesis being tested?

-Were correct statistical analysis used to support conclusions?

-Are there concerns about ethical or regulatory requirements being met?

Reviewer #1: (No Response)

Reviewer #2: The study's goals are explicitly stated and closely correspond to the research design. The selected population for the study is suitable for carrying out the research effectively, and the sample size is adequate to guarantee the statistical significance of the findings.

**Results**

-Does the analysis presented match the analysis plan?

-Are the results clearly and completely presented?

-Are the figures (Tables, Images) of sufficient quality for clarity?

Reviewer #1: (No Response)

Reviewer #2: The results are fine. The data analysis aligns seamlessly with the chosen methodology and analysis plan.

**Conclusions**

-Are the conclusions supported by the data presented?

-Are the limitations of analysis clearly described?

-Do the authors discuss how these data can be helpful to advance our understanding of the topic under study?

-Is public health relevance addressed?

Reviewer #1: (No Response)

Reviewer #2: The presented data substantiates the conclusion, and limitations are explicitly outlined. The manuscript effectively demonstrates the risk of leprosy transmission within households of leprosy cases, a crucial point, particularly in low-endemic settings.

**Editorial and Data Presentation Modifications?**

Reviewer #1: (No Response)

Reviewer #2: (No Response)

**Summary and General Comments**

Reviewer #1: The authors have adequately addressed the comments of this peer-reviewer.

Reviewer #2: The paper holds significant relevance, and the methodology is clearly explained, facilitating potential replication by other researchers in diverse settings. The study is valuable for comprehending the risk of contracting leprosy among household contacts of index cases. The paper identifies that those individuals living in the same household as the leprosy index case had a significant risk of contracting leprosy. This finding is especially relevant in a low-endemic settings like Senegal. The Proportion of new leprosy cases in this study is higher in households vs. beyond. Therefore, the focus of systematic contact trancing and SDR-PEP in countries with low endemicity of leprosy could be the household contacts or family members of the leprosy index cases. The high proportion of children among new leprosy cases shown by this stidy illustrates that there is a recent transmission. Thus, the strategy of active leprosy screening combined with the administration of a single dose of rifampicin as post-exposure prophylaxis used by the researchers is very necessary to interrupt further transmission. However, the results and findings of this study might not be applicable in countries like Brazil or India with high endemicity of leprosy.

PLOS authors have the option to publish the peer review history of their article (what does this mean?). If published, this will include your full peer review and any attached files.

Reviewer #1: No

Reviewer #2: No

---

## [Editor Report · Acceptance letter]

29 Jan 2024

Dear Dr. Batista,

We are delighted to inform you that your manuscript, "Feasibility of post-exposure-prophylaxis with single-dose rifampicin and identification of high prevalent clusters in villages’ hyperendemic for leprosy in Senegal," has been formally accepted for publication in PLOS Neglected Tropical Diseases.

Best regards,

Shaden Kamhawi

co-Editor-in-Chief

Paul Brindley

co-Editor-in-Chief
